# Co-Silencing of the Voltage-Gated Calcium Channel β Subunit and High-Voltage Activated α_1_ Subunit by dsRNA Soaking Resulted in Enhanced Defects in Locomotion, Stylet Thrusting, Chemotaxis, Protein Secretion, and Reproduction in *Ditylenchus destructor*

**DOI:** 10.3390/ijms23020784

**Published:** 2022-01-11

**Authors:** Mingwei An, Xueling Chen, Zhuhong Yang, Jianyu Zhou, Shan Ye, Zhong Ding

**Affiliations:** College of Plant Protection, Hunan Agricultural University, Changsha 410128, China; anmingwei@stu.hunau.edu.cn (M.A.); chenxueling@stu.hunau.edu.cn (X.C.); zhuhong_yang@hunau.edu.cn (Z.Y.); ldjyz6413@aliyun.com (J.Z.)

**Keywords:** plant-parasitic nematode, *Ditylenchus destructor*, voltage-gated calcium channels, Ca_v_β, RNAi

## Abstract

The voltage-gated calcium channel (VGCC) β subunit (Ca_v_β) protein is a kind of cytosolic auxiliary subunit that plays an important role in regulating the surface expression and gating characteristics of high-voltage-activated (HVA) calcium channels. *Ditylenchus destructor* is an important plant-parasitic nematode. In the present study, the putative Ca_v_β subunit gene of *D. destructor*, namely, *DdCa_v_β*, was subjected to molecular characterization. In situ hybridization assays showed that *DdCa_v_β* was expressed in all nematode tissues. Transcriptional analyses showed that *DdCa_v_β* was expressed during each developmental stage of *D. destructor*, and the highest expression level was recorded in the third-stage juveniles. The crucial role of *DdCa_v_β* was verified by dsRNA soaking-mediated RNA interference (RNAi). Silencing of *DdCa_v_β* or HVA *Ca_v_α_1_* alone and co-silencing of the *DdCa_v_β* and HVA *Ca_v_α_1_* genes resulted in defective locomotion, stylet thrusting, chemotaxis, protein secretion and reproduction in *D. destructor*. Co-silencing of the HVA Ca_v_α_1_ and Ca_v_β subunits showed stronger interference effects than single-gene silencing. This study provides insights for further study of VGCCs in plant-parasitic nematodes.

## 1. Introduction

Calcium channels are glycoproteins or lateral glycosylated protein complexes that are embedded in membrane lipids. These channels are widely present on the surface of the cell membrane and form a functional unit or pore. They have a high selective permeability for Ca^2+^ and control ion transport and cell function. Voltage-gated calcium channels (VGCCs) are important calcium channels that participate in many excitatory processes of cells, such as muscle contraction regulation, neurotransmitter release, gene regulation, and neuronal migration [1].

L-, N-, P/Q-, and R-type VGCCs consist of a central pore-forming α_1_ subunit and α_2_δ, β, and γ related auxiliary subunits [2,3]. The α_1_ subunit (Ca_v_α_1_) comprises of four homologous domains constituted by six putative transmembrane segments (S1′S6) [4,5]. The β subunit (Ca_v_β) is a cytosolic auxiliary subunit that binds to the proximal part of the intracellular loop between domains Ⅰ and Ⅱ of the α_1_ subunit [6,7]. Ca_v_α_1_ and Ca_v_β associate through a high-affinity interaction between the α-interaction domain (AID) and AID-binding pocket (ABP) [8].

Ca_v_β consists of five typical regions. The amino and carboxyl ends vary in both length and amino acid sequence. The Src homology 3 (SH3) and guanylate kinase (GK) domains are highly conserved and are connected by a weakly conserved HOOK domain, forming an N-SH3-HOOK-GK-C structure. The SH3-HOOK-GK core structure recapitulates many critical functions of Ca_v_β subunits, resulting in their classification as members of the membrane-associated GK protein family. Moreover, there are residues in the GK domain that interact with the AID [3,9,10]. As an auxiliary subunit, Ca_v_β has many functions; especially in regulating the surface expression and gating characteristics of high-voltage-activated (HVA) calcium channels [3,11]. Although Ca_v_α_1_ is the principal component of VGCCs, for the proper trafficking and functioning, the channels require Ca_v_β [12]. 

There are four subfamilies of Ca_v_βs (β1–β4) that are encoded by four distinct genes, each with splice variants, that have been identified in vertebrates. In *Caenorhabditis elegans*, two genes encoding β subunits, namely CCB-1 and CCB-2, were identified by genetic analysis and a homology search [13]. CCB-1 is necessary for viability [14]. However, there are few detailed reports on the Ca_v_β gene in plant-parasitic nematodes. Therefore, the study of Ca_v_β in *Ditylenchus destructor* will help us understand the role of VGCCs in plant-parasitic nematodes.

*D. destructor* develops and reproduces at 5–34 °C and each female lays 100–200 eggs in a lifetime [15]. It is easy to be cultured on fungi or potatoes and its life cycle is relatively short. Under suitable temperature and humidity conditions, the nematodes can complete a generation in 20 to 30 days [16]. *D. destructor* is one of the important pathogens of sweet potato (*Ipomoea batatas*) and causes significant yield loss of 20% to 50%, and up to 100% under heavy infestations [17,18]. In previous studies, HVA L-type (*DdCα1D*) and non-L-type (*DdCα1A*) VGCC α_1_ subunits of *D. destructor* were cloned, and their expression and the effects of their knockdown on nematode activity, including locomotion, chemotaxis, and reproduction, were studied [19]. Although some progress has been made in research on the function of Ca_v_α_1_ subunits in *D. destructor*, the function of HVA Ca_v_α_1_ in nematode stylet thrusting, protein secretion, and interaction with Ca_v_β remains unclear.

In the current study, we cloned the full-length cDNA encoding Ca_v_β of *D. destructor* and investigated the expression level and localization of Ca_v_β via qPCR and in situ hybridization. Using RNA interference (RNAi), we also further explored the effects of HVA Ca_v_α_1_ and Ca_v_β gene silencing on locomotion, chemotaxis, and reproduction of *D. destructor* and determined the effects of these genes on stylet thrusting and protein secretion. All the data that were generated provided additional clues to improve our understanding of VGCCs in *D. destructor*.

## 2. Results

### 2.1. Primary Structures of Ca_v_β of D. destructor

The full-length cDNA sequence of the *D. destructor* Ca_v_β, named *DdCa_v_β*, was obtained via PCR amplification (GenBank number: MN867027). The *DdCa_v_β* cDNA was 2036 bp long with a 171-bp 5′ untranslated region (UTR), a 1530-bp open reading frame (ORF), and a 335-bp 3′ UTR. The sequence was predicted to encode a protein of 509 amino acids, with a molecular mass of 56.88 kDa and a pI of 8.157. It contained an SL1 sequence at its 5’ end and a poly-A tail at its 3’ end (Figure 1A). The structure of *DdCa_v_β* protein was predicted by a voltage-dependent L-type calcium channel subunit beta-1 as a homology template, and the homology was 67.99% (Figure 1B).

*DdCa_v_β* was similar to the conventional β-subunit that is reported in vertebrates and invertebrates, including *C. elegans*, and contained five typical regions: amino (NH2) and carboxyl (COOH) termini with variable lengths and amino acid sequences, a highly conserved SH3 domain, a GK domain, and a variable and flexible HOOK region (Figure 1). As shown in Figure 1, *DdCa_v_β* had residues that were involved in interactions with the AID of Ca_v_α_1_. The *DdCa_v_β* protein was more homologous to nematode (*C. elegans*) CCB-1 (68.5%) and filaria (*Brugia malayi*) BMA-CCB-1 (67.5%) than to Ca_v_β in vertebrates (43.9–48.6%) (Table 1). *DdCa_v_β* showed 13.3% sequence identity with *C. elegans* CCB-2. 

### 2.2. Homology and Phylogenetic Analysis of DdCa_v_β

To investigate the evolutionary relationships of Ca_v_β, the neighbor-joining method in MEGA 7.0 software was used to construct phylogenetic trees comprising of the amino acid sequences of *DdCa_v_β* and other Ca_v_β protein sequences from invertebrates and vertebrates. As shown in the phylogenetic tree, conventional and variant β subunits were clustered into two large groups (Figure 2). The *D. destructor* Ca_v_β grouped with *C. elegans* CCB-1 and the *Loa loa*, *B. malayi*, *Drosophila melanogaster*, *Heterololigo bleekeri*, and *Lymnaea stagnalis* β subunits to form a large clade with conventional β-subunits that are found in many invertebrates. The Ca_v_β of vertebrates formed a large clade with the variant β subunits of *Homo sapiens* (β1–β4), *Oryctolagus cuniculus*, *Rattus norvegicus*, and *Xenopus laevis*. *C. elegans* CCB-2 formed a single, small clade. 

### 2.3. Tissue Localization of DdCa_v_β

To examine the site of *DdCa_v_β* mRNA expression in the tissue of *D. destructor*, in situ hybridization was performed. As shown in Figure 3, hybridization signals were detected in all the nematode tissues (Figure 3E–H), and hybridization with the control sense probe yielded no hybridization signals (Figure 3A–D). The wide distribution of *DdCa_v_β* indicates the distribution pattern of HVA calcium channels in *D. destructor*, as the β subunit is always connected to the HVA Ca_v_α_1_ subunit [3].

### 2.4. Expression of DdCa_v_β in Nematodes at Different Stages

The stage-specific expression of the *DdCa_v_β* transcript was analyzed via qPCR. *DdCa_v_β* was differentially expressed in *D. destructor* at different stages. With the expression level in second-stage juveniles (J2s) used as a standard, the expression of the β subunit gene was significantly upregulated from the egg stage to the third-stage juveniles (J3s) and then significantly downregulated in the fourth-stage juveniles (J4s). The expression in the J3 stage was the highest, and that in the J4 stage was the lowest (Figure 4).

### 2.5. Efficacy of Silencing HVA Ca_v_α_1_ and Ca_v_β Genes by dsRNA Soaking

To further explore the role of the HVA Ca_v_α_1_ and Ca_v_β, effective and specific silencing of these genes by RNAi was performed. We first evaluated the individual silencing of *DdCα1D*, *DdCα1A*, and *DdCa_v_β* and used a mixture of dsRNAs to target both *DdCα1D* or *DdCα1A*, and *DdCa_v_β* to evaluate the function of VGCCs.

Soaking with ds*Ca_v_β* specifically silenced the expression of *DdCa_v_β* by 47.0% compared to the control, while the expression of *DdCα1D* and *DdCα1A* was not changed (Figure 5A). Similar results were observed in the ds*Cα1D* and ds*Cα1A* treatments. The expression of *DdCα1D* and *DdCα1A* was reduced by 53.1% and 47.6%, respectively, compared to that in the ds*gfp* treatment, while the expression levels of other subunit genes were not changed (Figure 5B,C). When the nematodes were treated with a mixture of ds*Cα1D* and ds*Ca_v_β*, the relative expression levels of *DdCα1D* and *DdCa_v_β* were significantly downregulated by 57.4% and 50.9%, respectively, compared to those in the ds*gfp* treatment, while the expression of *DdCα1A* showed no significant change (Figure 5D). Under co-treatment with ds*Cα1A* and ds*Ca_v_β*, the relative expression levels of *DdCα1A* and *DdCa_v_β* were significantly downregulated by 49.8% and 55.2%, respectively, compared to those in the ds*gfp* treatment, while the expression of *DdCα1D* was not significantly changed (Figure 5E).

### 2.6. Analysis of the Knockdown Phenotype

#### 2.6.1. Observation of Locomotion Activity after RNAi

Nematode locomotion behaviors were tested via the migration assay in which the numbers of worms migrating through the columns and into the collection vials were counted at 6 h, 14 h, and 24 h. During the whole observation period, the rate of migration of nematodes that were treated with each target gene dsRNA was significantly lower than that of nematodes that were treated with ds*gfp*. The lowest nematode migration rates were found under the simultaneous treatment with ds*Cα1D* and ds*Ca_v_β*, with values of 12%, 33%, and 47% at 6 h, 14 h, and 24 h, respectively. In contrast, the nematode migration rates under ds*gfp* treatment at 6 h, 14 h, and 24 h were 32%, 65%, and 83%, respectively. The results showed that *DdCα1D* and *DdCa_v_β* could be important for the locomotion ability of *D. destructor* and that the simultaneous knockdown of both genes had a stronger effect on the locomotion ability (Figure 6).

#### 2.6.2. Attraction Rate Assay of *D. destructor* after RNAi

To evaluate the effect of knockdown of the α_1_ and β subunits on the chemotaxis of *D. destructor*, an in vitro chemotactic bioassay was carried out with sweet potato blocks in water agar plates. As shown in Figure 7, the attraction rate of nematodes that were treated with ds*Cα1D* and ds*Cα1D* + ds*Ca_v_β* decreased significantly to 7.13% and 3.88%, respectively, compared with 22.50% in the control. The silencing of the *DdCα1D* gene affected the chemotaxis of *D. destructor* toward the sweet potato, and this effect was more significant with co-silencing of the *DdCa_v_β* gene. 

#### 2.6.3. Stylet Thrusting of *D. destructor* after RNAi

The neurotransmitter serotonin is a conserved regulator of various behaviors in animals. The application of serotonin induces stylet thrusting in some plant-parasitic nematode species in the absence of a host [20,21,22]. To evaluate the effect of knockdown of the α_1_ and β subunits on the stylet thrusting of *D. destructor*, dsRNA-treated nematodes were soaked in serotonin (5 mM/L) for 20 min and the frequency of stylet thrusting per minute was observed under a microscope. As shown in Figure 8, the frequency of stylet thrusting per minute of the nematodes that were treated with ds*Cα1D* and ds*Cα1D* + ds*Ca_v_β* was significantly lower than that of the nematodes that were treated with ds*gfp*. The ds*Cα1D* + ds*Ca_v_β*-treated nematodes had the lowest frequency of stylet thrusting per minute at 13.5, compared to 53.0 for ds*gfp*-treated nematodes. The results showed that silencing the *DdCα1D* gene had a strong impact on the stylet thrusting of *D. destructor*, and this effect was more obvious with co-silencing of the *DdCa_v_β* gene.

#### 2.6.4. Detection of Secreted Proteins of *D. destructor* after RNAi

Based on pioneering work, resorcinol was used to induce esophageal gland secretion by *D. destructor* [23,24]. After soaking J3s in target dsRNAs and then incubating with 0.1% resorcinol for 16 h, the supernatant was used to detect the proteins that were secreted by the nematodes. For the worms that were treated with ds*Cα1A*, the secreted protein content was significantly reduced to 19,740 μg/mL compared to the controls that were treated with ds*gfp* (20,246.67 μg/mL). The secreted protein content of ds*Cα1A* + ds*Ca_v_β*-treated nematodes was the lowest at 19,280 μg/mL (Figure 9). The results showed that silencing of the *DdCα1A* gene had an effect on the protein secretion by *D. destructor*, and co-silencing of the *DdCa_v_β* gene enhanced this effect.

#### 2.6.5. Reproduction Rate of *D. destructor* after RNAi 

As shown in Figure 10, after 25 days, the reproduction rate of nematodes in each treatment group was significantly lower than that of nematodes that were treated with ds*gfp*. Compared with the reproduction rate of nematodes that were treated with ds*gfp* (79.1%), the reproduction rate of nematodes that were treated with both ds*Cα1A* and ds*Ca_v_β* was the lowest at only 13.6%. The results showed that gene silencing affected the reproduction of *D. destructor*, and simultaneous silencing of the *DdCα1A* and *DdCa_v_β* genes enhanced this effect.

## 3. Discussion

VGCCs are macromolecular complexes that are composed of a pore-forming α_1_ subunit and related α_2_/δ, β, and γ auxiliary subunits and are embedded in the plasma membrane of most excitable cells [2]. Ca_v_β is a kind of cellular solute auxiliary subunit that consists of five distinct regions, namely, the NH2 terminus, an SH3 domain, a HOOK region, a GK domain, and the COOH terminus. The NH2, HOOK, and COOH regions are variable in length and amino acid sequence, and the SH3 and GK domains are highly conserved [9,12]. In this study, a type of Ca_v_β was identified for the first time in *D. destructor*. Through analysis of the amino acid sequence of *DdCa_v_β*, we found that it has a typical β-domain. In addition, sequence alignment showed that its SH3 domain and GK domain are highly conserved. The phylogenetic tree indicated that *DdCa_v_β*, CCB-1, and the Ca_v_βs of *L. loa* and *B. malayi* were clustered together. According to these results, we preliminarily determined that the gene that was cloned in this study was the Ca_v_β of *D. destructor*.

*C. elegans* has two putative VGCC β subunits, namely, CCB-1 and CCB-2. We compared the amino acid sequences of CCB-1 and CCB-2 with the genome sequence of *D. destructor* in the WormBase database and identified a suspected β subtype [25]. After cloning, we found that the gene was highly homologous (68.48%) to CCB-1 and only 13.27% homologous to CCB-2. CCB-1, which is the main auxiliary subunit of the *C. elegans* VGCC, has contains two conserved subunits, namely, the SH3 and GK domains. CCB-1 can enhance the current of calcium channels, while CCB-2 does not affect this current and has no GK domain [12,14].

It has been reported that in the late growth stage of *C. elegans* and some plant-parasitic nematodes, the body wall muscle begins to undergo atrophy where the muscle mass and muscle function gradually decrease over time, and mobility gradually decreases [26,27]. In this study, *DdCa_v_β* was detected at different developmental stages and the expression level gradually increased from the egg stage to the J3 stage, followed by a significant decrease in the J4 stage. This may be related to atrophy of the body wall muscle in the later stage of stem nematode growth. In the early stage of nematode growth, the eggs and juveniles require more nutrients to complete the development and infection processes and body wall muscle mass and muscle function are continuously enhanced and improved. However, at the J4 stage, the body wall muscle of nematodes may begin to atrophy, and muscle mass and muscle function gradually decrease over time, leading to a decrease in *DdCa_v_β* gene expression.

Ca_v_β is expressed in muscle, neurons, and other tissues of vertebrates [12,28,29,30]. CCB-1 is widely expressed in most neuronal and muscle types in *C. elegans* [14]. Ye et al. showed that *DdCα1D* was expressed within body wall muscles and *DdCα1A* was expressed in the esophageal gland, vulva, and vas deferens of *D. destructor* [19]. In this study, we confirmed that *DdCa_v_β* was mainly expressed in the muscles. The results are similar to those regarding the expression of Ca_v_β in vertebrates and *C. elegans*, and as the main auxiliary subunit, it is also consistent with the results of Ye et al.

Ca_v_α_1_ is the main functional subunit of VGCCs and plays an important role in the pharyngeal and body wall muscles of *C. elegans* [31,32,33]. Ye et al., by RNAi, confirmed that the *DdCα1D* gene plays a key role in the modulation of the cell wall muscle and normal locomotion in *D. destructor* and that the *DdCα1A* gene plays an important role in reproduction regulation in nematodes [19]. In this study, we silenced the HVA Ca_v_α_1_ and β subunits, individually and in combination, to examine the effects on the locomotion behaviors and reproduction rate of the nematodes. The results showed that the locomotion ability of *D. destructor* was greatly affected by *DdCα1D* gene silencing, and that the reproduction rate of *D. destructor* was affected by silencing of the *DdCα1A* gene, especially when the *DdCa_v_β* gene and α_1_ subunit were co-silenced. It is suggested that the Ca_v_α_1_ subunit plays an important role in the movement and reproduction of *D. destructor*. As the main auxiliary subunit, the Ca_v_β subunit plays an important auxiliary role with Ca_v_α_1_. At the same time, these results further confirmed the functions of the L-type and non-L-type α_1_ subunits in the movement and reproduction of *D. destructor*.

Chemotaxis is movement in the direction of higher concentrations of semichemicals, such as plant chemical signals [34]. Chemical sensing by nematodes is an important part of their host-seeking behavior. In chemosensory neurons, odor concentration information is determined by the time integration of the increased intracellular calcium concentration of L-type VGCCs in a pair of olfactory neurons [35]. To date, only egl-19 (L-form) and unc-2 (non-L-form) have been shown to affect the chemical sensory signals of *C. elegans* [31,36,37]. Among the olfactory neurons of *C. elegans*, AWA neurons can guide nematodes to find potential food sources through unstable signals that are produced by bacteria and activate calcium-mediated action potentials that are initiated by egl-19 [38,39,40]. In our study, nematode chemotaxis was significantly inhibited by *DdCα1D* (L-type) gene silencing, especially with co-silencing of the *DdCa_v_β* gene. However, when the *DdCα1A* (non-L-type) gene was silenced, the chemotactic inhibition effect on the nematode was not as obvious as that which was observed upon silencing of the *DdCα1D* gene. The results showed that the L-type and non-L-type α_1_ subunits affected the chemotaxis of *D. destructor* toward sweet potato, and the L-type α_1_ had a strong effect on *D. destructor* movement. The results are similar to those for *C. elegans*. At the same time, the results show that the β subunit, as the main auxiliary subunit of voltage-gated ion channels, plays an important auxiliary role with the α_1_ subunit.

Serotonin is a neuroregulator that regulates feeding behavior in almost all phyla of the animal kingdom [41]. It also acts as a neuroregulator in *C. elegans* and is related to physiological functions such as pharyngeal pumping, egg laying, and locomotion [41,42,43]. It has been shown that, in *C. elegans*, serotonin activates pharyngeal pumping by the SER-7 serotonin receptor (a G protein-coupled receptor) in MC motor neurons, and SER-7 activates the downstream Gsα signaling pathway and then stimulates cholinergic transmission from MCs to the pharyngeal muscle [44]. To stimulate MC motor neurons, the CCA-1 (T-type) channel is involved in the initiation of action potentials, helping the membrane reach the threshold for activating EGL-19 (L-type) channels after the excitatory postsynaptic potential of the MC motoneurons and allowing reliable and rapid depolarization and contraction of the pharyngeal muscle [45]. In this study, soaking the treated nematodes with serotonin at 5 mM for 20 min significantly decreased the frequency of stylet thrusting of the nematodes after silencing of the *DdCα1D* gene, especially with co-silencing of the *DdCa_v_β* gene. The frequency of needle twitching of the nematodes did not change significantly after *DdCα1A* gene knockdown. In addition, *D. destructor* and *C. elegans* do not have voltage-gated Na^+^ channels, and the action potential depends on VGCCs [46,47]. In the pharyngeal muscles, the L-type VGCCs contribute to shaping the action polarization phase [48]. This suggests that the L-type α_1_ subunit plays a key role in the contraction of *D. destructor* stylet muscles. 

In plant-parasitic nematodes, proteins that are synthesized in the esophageal gland (subventral and dorsal glands) and secreted through the stylet play important roles in the host-nematode relationship. By detection of the secreted nematode proteins after gene silencing, we found that the *DdCα1A* gene had a strong effect on protein secretion by *D. destructor*, especially when the *DdCα1A* and *DdCa_v_β* genes were co-silenced. We speculate that the non-L-type α_1_ subunit plays an important role in regulating protein secretion in the esophageal glands of *D. destructor*. At the same time, this result not only echoed the results of the expression localization study of the non-L-type α_1_ subunit in our previous work but also indicated the important auxiliary role of the β subunit with the α_1_ subunit.

## 4. Materials and Methods

### 4.1. Nematode Culture

*D. destructor*, isolated from diseased sweet potato and cultured on potato dextrose agar (PDA) plates that were inoculated with *Fusarium semitectum*, was employed in this study. After 30 days of incubation at 25 °C, *D. destructor* worms in mixed life stages were washed off the plates with distilled water and collected by the Baermann funnel technique [49]. The nematode eggs were collected via density gradient centrifugation, inoculated on PDA plates, and collected every 7 days to obtain nematodes at different stages [50].

### 4.2. RNA Extraction

Total RNA was extracted from *D. destructor* in mixed stages and different individual life stages by using TRIzol reagent (Invitrogen, Waltham, MA, USA) according to the manufacturer’s instructions. First-strand cDNA was synthesized from the total RNA by using the Super-Script Ⅲ First-Strand Synthesis Kit (Invitrogen) with OligodT primers according to the manufacturer’s protocol.

### 4.3. Cloning the Ca_v_β of D. destructor

On the basis of the protein sequence of Ca_v_β of *C. elegans*, the putative Ca_v_β gene sequence of *D. destructor* was identified in the WormBase database (https://parasite.wormbase.org/index.html, accessed on 24 November 2021), and primers (MF, MR) were designed for cloning the conserved sequence (Table 2). The PCR conditions used were as follows: 94 °C for 5 min; 30 cycles of 94 °C for 30 s, 48 °C for 30 s and 72 °C for 2 min; and a final extension at 72 °C for 7 min. The 3′-end cDNAs of the β subunit gene were obtained by rapid amplification of cDNA ends (RACE) PCR in conjunction with the use of a PrimeScript RTase Kit (TaKaRa, Japan). The 3′ RACE outer primer and 3′ RACE inner primer were provided with the kits. The 5′-end cDNAs of the β subunit gene were amplified with a primer upstream of the spliced leader (SL1) sequence, which is a sequence that *is* specific to nematode mRNA [51]. Gene-specific primers (3′ β-F1, 3′ β-F2, 5′ β-R1, and 5′ β-R2) (Table 2) were designed for 3′ and 5′ RACE amplification based on the conserved sequence of the β subunit gene; these sequences were obtained from previous sequencing results. After initial confirmation via agarose gel electrophoresis, all the resulting nested PCR products were cloned into the pMD™ 19-T vector (TaKaRa, Japan), and a single colony was sequenced (Sangon Biotech Co., Ltd., Shanghai, China). 

### 4.4. Gene Expression Analysis by qPCR

qRT-PCR was used to assess the gene expression patterns of Ca_v_β. The total RNA was isolated from nematodes at different developmental stages (egg, J2, J3, and J4). The concentration of each sample was analyzed by a microspectrophotometer, and the samples were diluted to the same concentration with ddH_2_O. The RNA was then reverse-transcribed to cDNA and the relative gene expression of Ca_v_β was quantified via qRT-PCR. 18S rRNA was used as an endogenous control and the primers that were used are listed in Table 2. qRT-PCR was performed using SYBR Green mix according to the manufacturer’s instructions (Bio-Rad, Hercules, CA, USA). The qRT-PCR conditions were as follows: 95 °C for 2 min, followed by 39 cycles of 95 °C for 5 s and 59 °C for 40 s. qRT-PCR data from three biological and technical replicates were analyzed using Bio-Rad CFX Manager™ software. The relative transcript levels of each sample were calculated using the 2^−∆∆Ct^ method [52]. The experiment included three biological repeats and three technical repetitions.

### 4.5. In Situ Hybridization of Ca_v_β in D. destructor

DIG-labeled sense and antisense probes were amplified via DIG RNA Labeling Mix (Roche Applied Science, Penzberg, Germany) together with BamHIF- and HindIIIR-specific primers (Table 2). In situ hybridization was performed as previously described [53], with some modifications. Briefly, *D. destructor* nematodes were fixed in 4% paraformaldehyde at 4 °C for 18 h, followed by an additional incubation at 22 °C for 4 h. Hybridization signals in the nematodes were detected with alkaline phosphatase-conjugated anti-DIG antibodies and substrate (Sangon Biotech Co., Ltd., Shanghai, China). Finally, the nematodes were placed on slides and then observed and imaged under a microscope (Carl Zeiss, Germany).

### 4.6. In Vitro RNA Interference Targeting VGCCs in D. destructor 

Total *D. destructor* cDNA was used as a template for double-stranded RNA (dsRNA) synthesis using the MEGA Script RNAi Kit (Ambion, Austin, TX, USA) according to the manufacturer’s protocol. T7-labeled gene-specific primers (Table 2) were designed to amplify regions of the Ca_v_β and Ca_v_α_1_ (L-type and non-L-type α_1_ subunit) transcripts. A nonendogenous control GFP dsRNA was synthesized using the GFP-F and GFP-R primers (Table 2). An RNAi assay for nematodes was conducted as described previously [54]. Approximately 10,000 J3-stage nematodes were soaked in 0.4 μg/μL target dsRNA in 200 μL of soaking buffer for 24 h in the dark on a slowly moving rotator at room temperature. In addition, J3s that were incubated in ds*gfp* and in soaking buffer (without dsRNA) served as the control. The nematodes that were treated with dsRNA were washed three times with ddH_2_O, RNA was extracted, and qPCR was performed using the methods that are described above to analyze the suppression of Ca_v_β and Ca_v_α_1_ mRNA expression in *D. destructor*. FITC was used to trace the efficacy of the uptake of dsRNA by *D. destructor*. This experiment was independently repeated three times.

### 4.7. Analysis of Knockdown Phenotypes

After soaking for 24 h, the third-stage juveniles (J3s) were collected by centrifugation at 5000× *g* for 2 min and washed 3 times with ddH_2_O. The effects of Ca_v_β RNAi on the locomotion, chemotaxis, and reproduction rates of *D. destructor* were evaluated by the method of Shan Ye et al. [19]. In brief, worm motility was evaluated using a sand column functional migration assay. Approximately 100 dsRNA-treated J3s were added to the top of moistened sand columns and the columns were placed vertically in collection vials. The numbers of worms that were migrating through the columns and into the collection vials were counted at 6 h, 14 h, and 24 h. A total of 100 dsRNA-treated worms were inoculated on PDA with *Fusarium semitectum*. After 25 days, the nematodes were isolated and collected from the Petri dish and the total number of reproductions was counted to calculate the reproduction rate. The chemotaxis test was carried out on a 1% water agar plate in 90 mm glass dishes. Approximately 200 treated and control nematodes were added to the circular hole (1 cm diameter) in the center of the plate. Then, sweet potato slices with a diameter of 1 cm were placed 3.5 cm from the center, and the Petri dish was sealed with fresh-keeping film. After being placed in a dark incubator at 25 °C for 36 h, the nematodes within 2 cm of the sweet potato chips were isolated and counted and the attraction rate of sweet potatoes for nematodes was calculated. There were three biological repeats and three technical repeats in each detection experiment. The attraction rate was calculated as follows [55]: Attraction rate (%) = Number of nematodes induced/Total number of nematodes input × 100.

The bioassay for nematode stylet thrusting was adapted from methodology that was described previously [56]. Approximately 50 J3s were incubated with 20 µL of serotonin solution (5 mM/L) in a 1.5 mL centrifuge tube for 20 min. The frequency of stylet thrusting per minute was observed under a microscope. A total of 10 nematodes were selected for each treatment. To detect the protein that was secreted by the esophageal gland of the nematode, approximately 2000 J3s were added to a 1.5 mL centrifuge tube containing 100 mL of 0.1% resorcinol neurotransmitter solution. After incubation at 25 °C for 16 h, the supernatant was used for determination of the secreted protein content with a Modified BCA Protein Assay Kit (Sangon Biotech Co., Ltd., Shanghai, China). Each process was performed for three biological repeats and three technical repeats.

### 4.8. Bioinformatic Analysis

To search for homologs, Ca_v_β sequences were compared using BLAST against the NCBI database (https://blast.ncbi.nlm.nih.gov/Blast.cgi, accessed on 12 October 2020). The Ca_v_β sequences were aligned using the DNAMAN software package (version 5.2.2, Lynnon Biosoft, San Ramon, CA, USA). Phylogenetic analyses of the Ca_v_β amino acid sequences and homologous sequences were obtained from the NCBI database (http://www.ncbi.nlm.nih.gov/blast/Blast.cgi, accessed on 12 October 2020) and prediction of its protein tertiary structure using the SWISS-MODEL online tool (https://www.swissmodel.expasy.org/, accessed on 24 November 2021).

## 5. Conclusions

Together, our results indicate that the Ca_v_β subunit plays an important auxiliary role associated with the Ca_v_α_1_ subunit in *D. destructor* and further validates the role of the L-type and non-L-type α_1_ subunits in *D. destructor*. All the data provide additional clues for improving our understanding of the VGCCs of *D. destructor*.

## Figures and Tables

**Figure 1 ijms-23-00784-f001:**
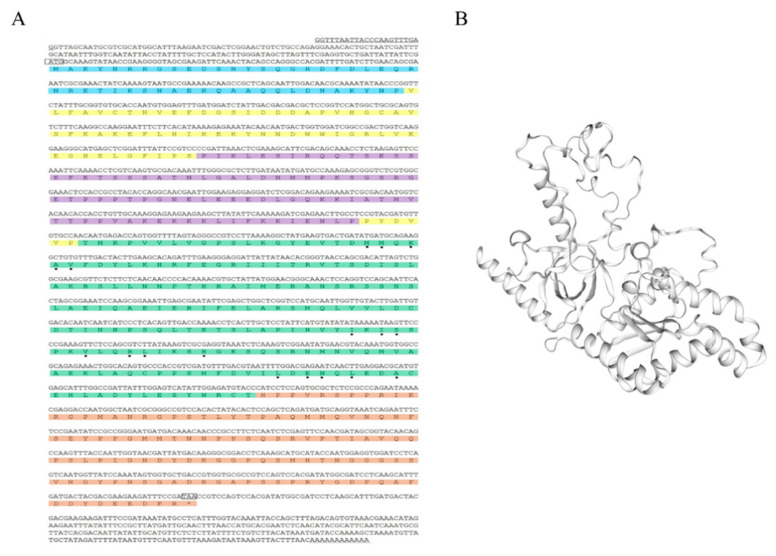
*DdCa_v_β* sequence analysis. (**A**), The *DdCa_v_β* cDNA sequence and deduced amino acid sequence. The start codon (ATG) and the stop codon (TAA) are indicated in boxes. The fragment that is not translated before the start codon is the 3’ UTR, and the fragment that is not translated after the termination codon is the 5’ UTR. The underlined area represents SL1 at the front end of the 5’ UTR and the poly-A tail at the end of the 3’ UTR. The asterisk (*) indicates the stop codon (TAA). The light blue and brown colors indicate the NH_2_ terminus and the COOH terminus, respectively; yellow, the SH3 domain; purple, the HOOK region; and green, the GK domain. The residues that are involved in interactions with the AID are marked with the “•” symbol. (**B**), The predictive structure of *DdCa_v_β*.

**Figure 2 ijms-23-00784-f002:**
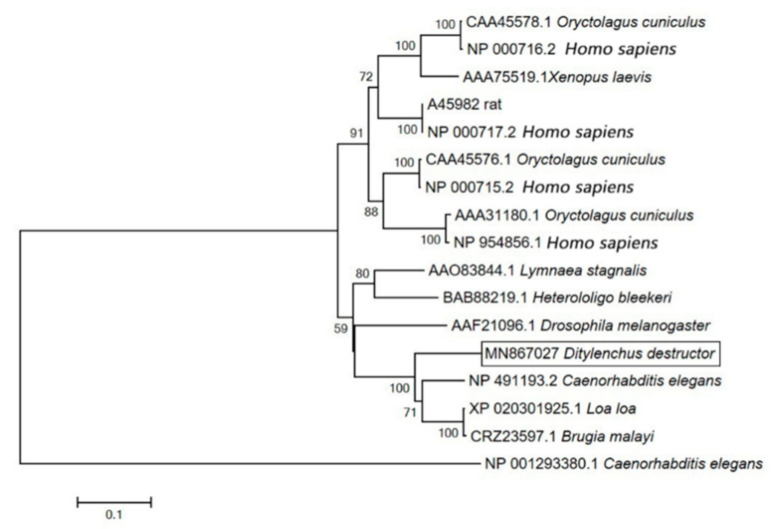
Molecular phylogenetic analysis of *Dd**Ca_v_**β* via the neighbor-joining method. The species that were used in phylogenetic tree construction are summarized in Table 1. The phylogram was constructed based on the amino acid sequences of 16 Ca_v_β proteins using MEGA 7.0. The numbers below the branches indicate the bootstrap values.

**Figure 3 ijms-23-00784-f003:**
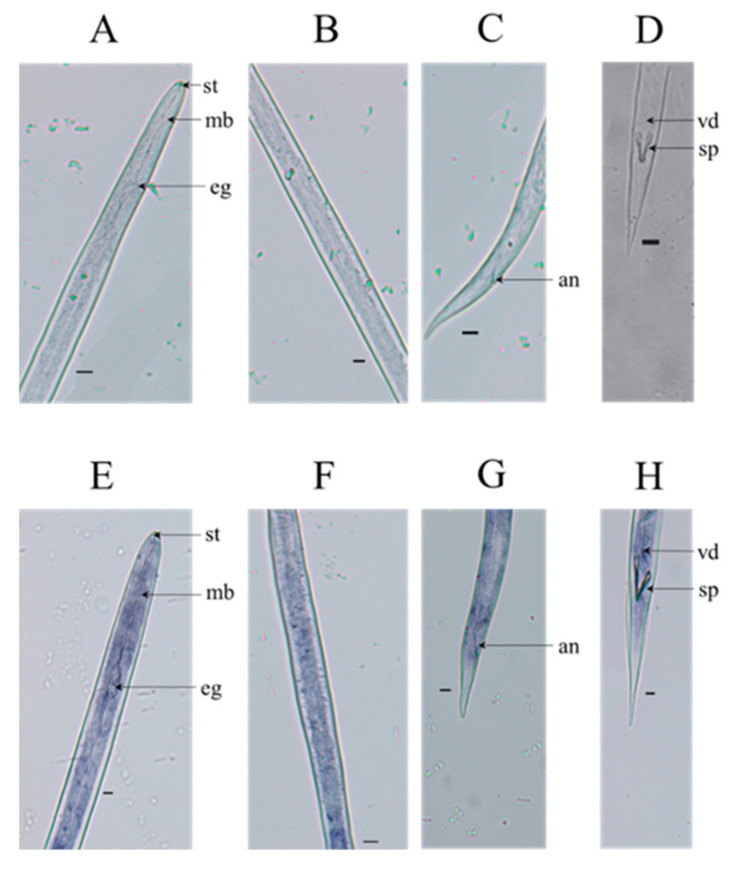
Localization of the expressed *Dd**Ca_v_**β* gene in *D. destructor* via in situ hybridization. (**A**–**D**) In situ hybridization of sense probes for *Dd**Ca_v_**β* in *D. destructor*. (**E**–**H**) In situ hybridization of antisense probes for *Dd**Ca_v_**β* in *D. destructor*. st: stylet; mb: median bulb; e.g., esophageal gland; an: anus; vd: vas deferens; sp: spicules. Scale bar = 20 μm.

**Figure 4 ijms-23-00784-f004:**
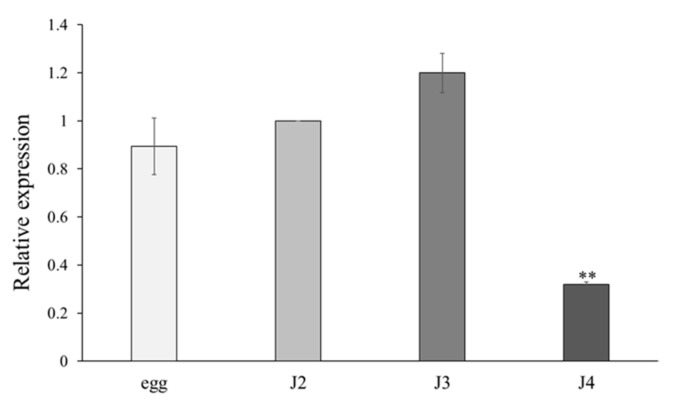
The relative expression levels of *Dd**Ca_v_**β* at different developmental stages of *D. destructor*. The expression level in J2s was used as the standard value. The data are presented as the mean ± s.d of three biological replicates and three technical replicates (*n* = 9). Asterisks indicate a significant difference at the level of *p* < 0.01, as tested by Duncan’s new complex difference method.

**Figure 5 ijms-23-00784-f005:**
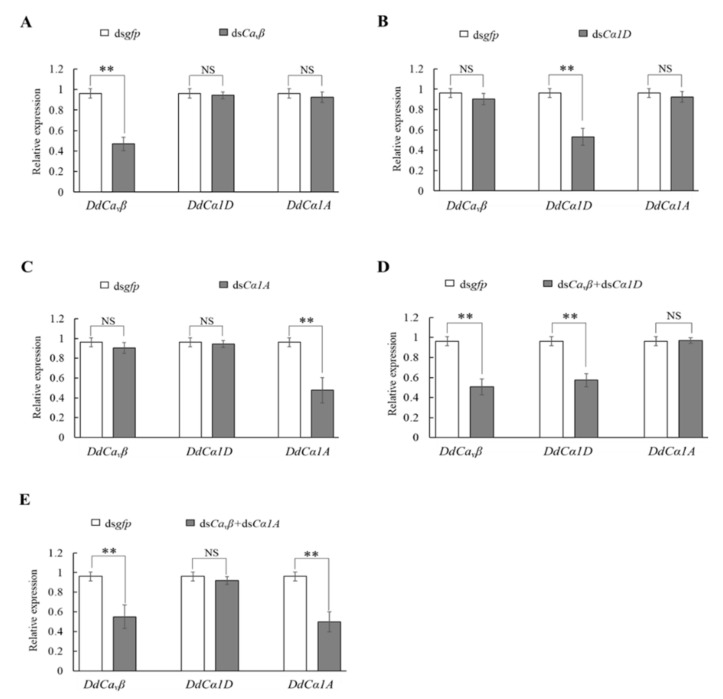
The relative expression levels of *Dd**Ca_v_**β*, *DdCα1D*, and *DdCα1A* in *D. destructor* after soaking with specific dsRNA. (**A**) The relative expression levels of the *Dd**Ca_v_**β*, *DdCα1D*, and *DdCα1A* genes after ds*Ca_v_**β* treatment. (**B**) The relative expression levels of the *Dd**Ca_v_**β*, *DdCα1D*, and *DdCα1A* genes after d*sCα1D* treatment; (**C**) The relative expression levels of the *Dd**Ca_v_**β*, *DdCα1D*, and *DdCα1A* genes after ds*Cα1A* treatment; (**D**) The relative expression levels of the *Dd**Ca_v_**β*, *DdCα1D*, and *DdCα1A* genes after ds*Cα1D* + ds*Ca_v_**β* treatment; (**E**) The relative expression levels of the *Dd**Ca_v_**β*, *DdCα1D*, and *DdCα1A* genes after ds*Cα1A* + ds*Ca_v_**β* treatment. Significant differences between the treatment and control are indicated with a line with asterisks (** *p* < 0.01; Student’s *t* test). “NS” indicates that there was no significant difference between the samples.

**Figure 6 ijms-23-00784-f006:**
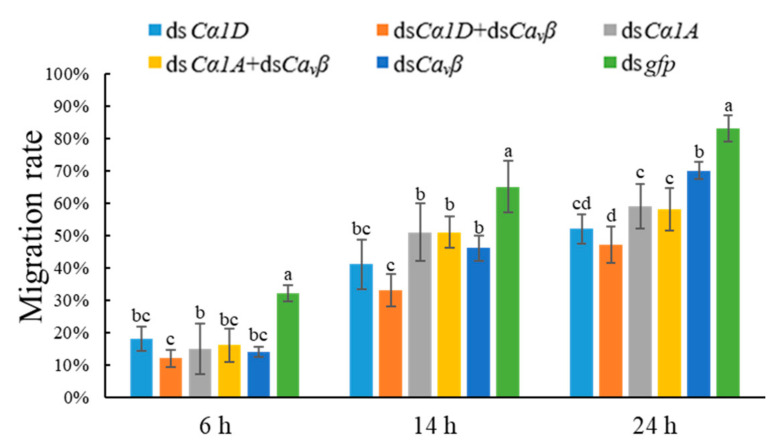
Silencing of *Ca_v_**β* or HVA *Ca_v_**α_1_* and co-silencing of *Ca_v_**β* and HVA *Ca_v_**α_1_* affected the locomotory activity of *D. destructor*. Nematodes that were treated with *gfp* dsRNA were used as controls. Different lowercase letters indicate a significant difference at the level of *p* < 0.05, as tested by Duncan’s new complex difference method.

**Figure 7 ijms-23-00784-f007:**
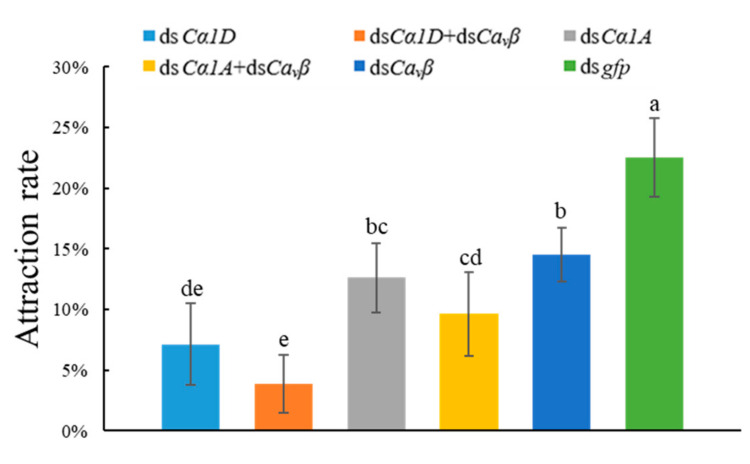
Silencing of *Ca_v_**β* or HVA *Ca_v_**α_1_* and co-silencing of *Ca_v_**β* and HVA *Ca_v_**α_1_* caused defects in the chemotaxis of *D. destructor* toward sweet potato slices. Nematodes that were treated with *gfp* dsRNA were used as controls. Different lowercase letters indicate a significant difference at the level of *p* < 0.05, as tested by Duncan’s new complex difference method.

**Figure 8 ijms-23-00784-f008:**
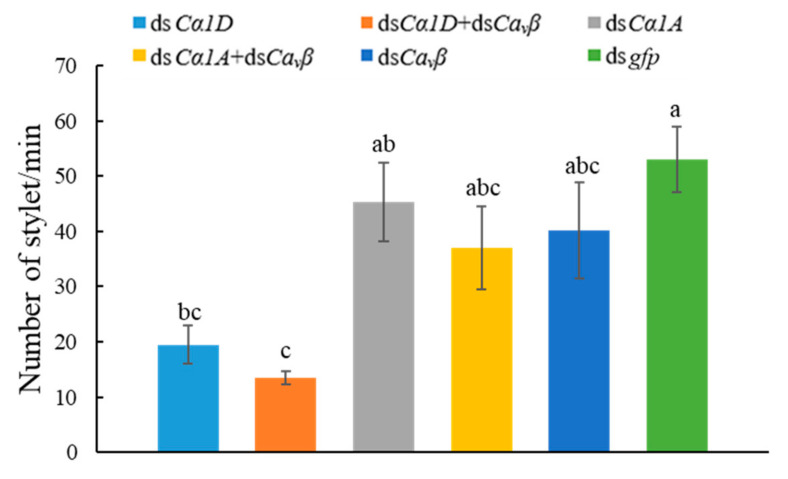
Silencing of *Ca_v_**β* or HVA *Ca_v_**α_1_* and co-silencing of *Ca_v_**β* and HVA *Ca_v_**α_1_* affected the stylet thrusting of *D. destructor*. Nematodes that were treated with *gfp* dsRNA were used as controls. Different lowercase letters indicate a significant difference at the level of *p* < 0.05, as tested by Duncan’s new complex difference method.

**Figure 9 ijms-23-00784-f009:**
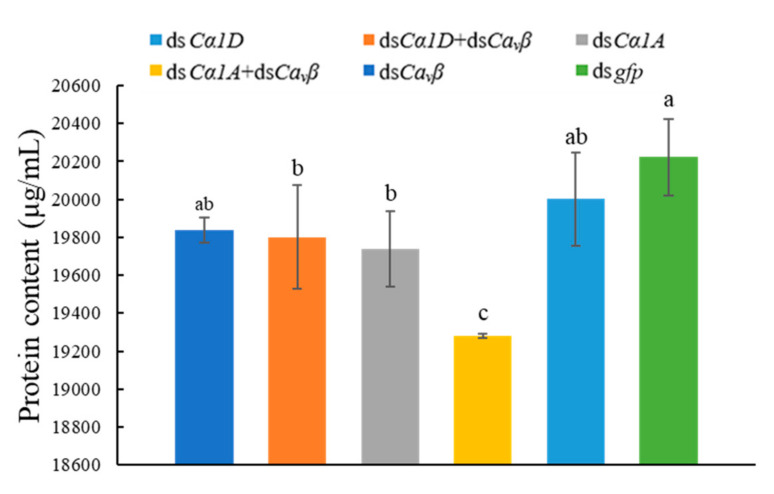
Silencing of *Ca_v_**β* or HVA *Ca_v_**α_1_* and co-silencing of *Ca_v_**β* and HVA *Ca_v_**α_1_* affected the protein secretion by *D. destructor*. Nematodes that were treated with *gfp* dsRNA were used as controls. Different lowercase letters indicate a significant difference at the level of *p* < 0.05, as tested by Duncan’s new complex difference method.

**Figure 10 ijms-23-00784-f010:**
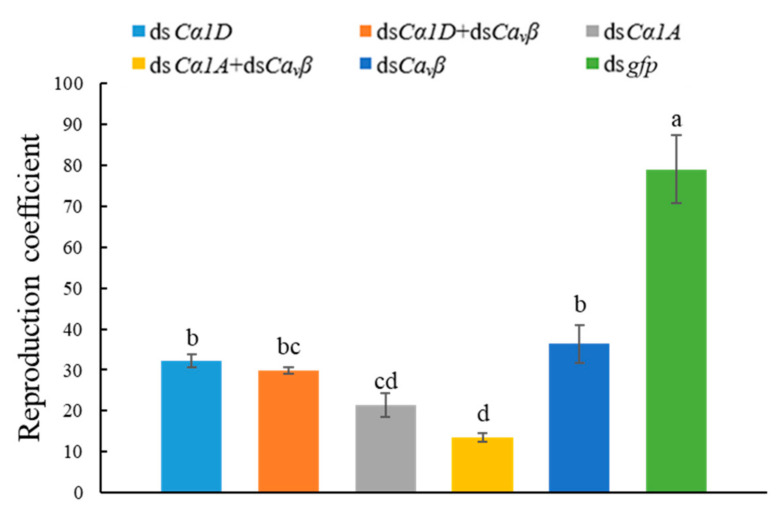
Silencing of *Ca_v_**β* or HVA *Ca_v_**α_1_* and co-silencing of *Ca_v_**β* and HVA *Ca_v_**α_1_* affected the reproduction rate of *D. destructor*. Nematodes that were treated with *gfp* dsRNA were used as controls. Different lowercase letters indicate a significant difference at the level of *p* < 0.05, as tested by Duncan’s new complex difference method.

**Table 1 ijms-23-00784-t001:** Ca_v_β information used for sequence alignment and phylogenic analysis.

Species	Molecule Name/Accession Number	Identity (%)
*Brugia malayi*	BMA-CCB-1/CRZ23597.1	67.5
*Loa loa*	beta 2a/XP_020301925.1	67.2
*Caenorhabditis elegans*	CCB-1/NP_491193.2	68.5
	CCB-2/NP_001293380.1	13.3
*Ditylenchus destructor*	MN867027	
*Drosophila melanogaster*	AAF21096.1	49.2
*Heterololigo bleekeri*	BAB88219.1	50.2
*Lymnaea stagnalis*	AAO83844.1	50.5
*Homo sapiens*	beta-1/NP_954856.1	44.8
	beta-2/NP_000715.2	46.5
	beta3/NP_000716.2	47.92
	beta-4/NP_000717.2	48.7
*Oryctolagus cuniculus*	beta-subunit/AAA31180.1	44.9
	CaB2b/CAA45576.1	43.9
	CaB3/CAA45578.1	46.6
*Rattus norvegicus*	beta4/A45982	48.6
*Xenopus laevis*	AAA75519.1	46.4

**Table 2 ijms-23-00784-t002:** Polymerase chain reaction (PCR) primers.

Name of Primer	Sequence (5′-3′)	Purpose
MF	GATGGCAAAGTATAACCGAAG	Primers used for *DdCa_v_β* cloning
MR	TTATCGGAAATCTTCTTCGTCGTAG	
3′ RACE outer primer	TACCGTCGTTCCACTAGTGATTT	
3′ RACE inner primer	CGCGGATCCTCCACTAGTGATTTCACTATAGG	
3′ β-F1	ACCTCACTTGCTCCTATTCATGTAT	
3′ β-F2	CCGTCCAGTCCACGATATGGCGAT	
SL1	GGTTTAATTACCCAAGTTTGAG	
5′ β-R1	TTTGGAGCTCTTAGAGGTTTG	
5′ β-R2	TTAGAGGTTTGCTGTCGAATG	
18SF	CTGATTAGCGATTCTTACGGA	Primers for real-time PCR analysis
18SR	AGAAGCATGCCACCTTTGA	
q-β-F	AGCCGCTCAGCAATTGGACA	
q-β-R	TGAAAGACACTGCGCAGCCA	
q-L-F	GACCCGTTATTGTTGAGCCA	
q-L-R	ACGTTCCTTCGAGATGAGA	
q-NL-F	TAGAAAACAGGCGAGACTTCC	
q-NL-R	CTCATCCGTTGTTCGATCCTC	
BamHI F	CGGGATCCGGAACGAGCAAACTCCAGGTC	Primers for ISH analysis
HindⅢ R	CCCAAGCTTATGCTCACATGCGTCCTCAAG	
dβ-F	TAATACGACTCACTATAGGGAAGTTCCCCGAAAGTTCTCCAG	Primers used for synthesizing dsRNA
dβ-R	TAATACGACTCACTATAGGGAGGTCCGCCCTTGTCATAATC	
dL-F	TAATACGACTCACTATAGGGAGGAAGATGACCTCTTGTTAG	
dL-R	TAATACGACTCACTATAGGGCCCAATATATGACCGTCTTTG	
dNL-F	TAATACGACTCACTATAGGGCGCAACACGTACCAAACTC	
dNL-R	TAATACGACTCACTATAGGGCTCATCTGAATCGCTAAGAGG	
GFP-F	TAATACGACTCACTATAGGGTACATCGCTCTTTCTTCACCG	
GFP-R	TAATACGACTCACTATAGGGACCAACAAGATGAAGAGCACC	

The restriction enzyme sites and the T7 promoter sequences are underlined.

## Data Availability

Not applicable.

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
