# Peer review of "Co-Silencing of the Voltage-Gated Calcium Channel β Subunit and High-Voltage Activated α1 Subunit by dsRNA Soaking Resulted in Enhanced Defects in Locomotion, Stylet Thrusting, Chemotaxis, Protein Secretion, and Reproduction in Ditylenchus destructor"

_ijms, 2022, doi:10.3390/ijms23020784_

Round 1

Reviewer 1 Report

In the present manuscript, the authors cloned the full-length cDNA encoding Cavβ of Ditylenchus destructor and investigated the expression level and localization of DdCavβ via qPCR and in situ hybridization. Using RNA interference (RNAi), they also further explored the effects of HVA Cavα1 and Cavβ gene silencing on locomotion, chemotaxis and reproduction of D. destructor and determined the effects of these genes on stylet thrusting and protein secretion. In my opinion, this manuscript can be considered for publication after minor revision according to the following suggestions:

  • The authors should include a short paragraph describing the life cycle of the parasite to help readers less familiar with this plant-parasitic nematode.

  • Observation of locomotion activity and reproduction rate of D. destructor after RNAi is not described in Material and methods.

  • Lines 194, 202 and 344: “knockout” change to “knockdown”.

  • Scientific names do not always appear in italics in the manuscript.

  • There is a lot of information repeated throughout the manuscript. Thus, for example, the same information can be read many times in Material and methods, Results, and figure legends. Authors should make an effort to condense this information and not be so repetitive.

Reviewer 2 Report

In this study the authors cloned the full-length Cav-beta subunit from plant-parasitic nematode Ditylenchus destructor, and identified some of its characteristics within this species, giving guidance for further study against this kind of parasite which could be useful in agricultural practices. 

  The study in this manuscript systematically depicted the role of Cav-beta in regulating the behavior of D. destructor. However, the function of a protein is tightly connected with its structure. Cav-beta regulates the function of VGCC to control the behavior of D. destructor not only by its amino acid sequence but also by its tertiary structure. Given the advantages of structure prediction and determination tools. I suggest the authors make a minor change as follow:

        In line 45 the authors mentioned that Cav-beta have five typical regions and they are highlighted as amino acid sequence in figure 1. From this information, the readers may easily raise the question of what the structure of Cav-beta is. Here, my suggestion is to add a figure with the structure of Cav-beta, which could be either the topology distribution from its homologues with known structure, or the predicted structure from AlphaFold2 or some other structural prediction software. Such a minor change could better help people vividly understand the function from different part of Cav-beta that was ​described in this manuscript.
